# The Role of Parental Communication and Emotional Intelligence in Child-to-Parent Violence

**DOI:** 10.3390/bs9120148

**Published:** 2019-12-09

**Authors:** Paula López-Martínez, David Montero-Montero, David Moreno-Ruiz, Belén Martínez-Ferrer

**Affiliations:** 1Department of Education and Social Psychology, Pablo Olavide University, 41013 Seville, Spain; dmonmon@alu.upo.es (D.M.-M.); bmarfer2@upo.es (B.M.-F.); 2Department of Social Psychology, Valencia University, 46003 Valencia, Spain; david.moreno-ruiz@uv.es

**Keywords:** child-to-parent violence, family communication, emotional intelligence, adolescence

## Abstract

In recent years, cases of child-to-parent violence (CPV) have increased significantly, prompting greater scientific interest in clarifying its causes. The aim of this research was to study the relationship between styles of family communication (open, offensive and avoidant), emotional intelligence or EI (attention, repair and perceived emotional clarity) and CPV, taking into account the gender of the aggressors. The participants of the study were 1200 adolescents (46.86% boys) between the ages of 12 and 18 enrolled at secondary schools in the Autonomous Communities of Andalusia and Valencia (M = 13.88, SD = 1.32). A Multivariate Analysis of Variance (MANOVA, 3 × 2) was performed with CPV and gender as independent variables and family communication styles and EI as dependent variables. The results showed that the adolescents with low CPV obtained lower scores for offensive and avoidant family communication and higher scores for both positive family communication and emotional repair. The girls scored higher than the boys in both offensive communication and perceived emotional attention. The results highlight the importance of encouraging positive communication, as well as the need to strengthen perceived emotional repair to prevent future cases of CPV.

## 1. Introduction

In recent years, the number of complaints filed by parents against their children for violence towards them has increased by 88.52%, from 2683 complaints in 2007 to 5058 in 2018, the last year for which figures are available according to data from the Prosecutor’s Office [1]. Child-to-parent violence (CPV) represents a growing social problem. There are several factors that may explain the increase in child-to-parent violence, such as: family, individual, socio-cultural and educational. In Spain, the age of becoming a father is growing, which has given rise to elderly parents, with less energy to maintain discipline and set limits. In addition, difficulties in reconciling work and family life have led children to spend more time without parental supervision. Parents have little time to be with them, so they tend to avoid tension conflict and establish less parental control. It is common for parents to attribute their children’s educational responsibilities to different institutions, such as schools. However, when other adults, such as teachers, try to put limits, parents often ally with their children and defend them, so that the link between school and family weakens. Finally, society has evolved into an educational model based on reward, where conflict resolution through violence is increasingly common [2,3]. Victims of child-to-parent violence are often unaware of its existence as they underestimate the violence suffered, hence the actual figure is properly higher than recorded [4,5]. Some parents may feel uncomfortable admitting that their sons and daughters treat them aggressively, since society often interprets that CPV implies a failure on the part of fathers and mothers in educating and setting limits for their children [6].

More specifically, CPV is defined as any harmful act committed by a teenager with the intention of obtaining power and control over either of his or her parents [7]. A more current definition specifies the use of physical, psycho-emotional and/or economic violence repeatedly and over time, in order to dominate and control parents or those exercising parental functions [8]. It is common for CPV to start with the economic form before progressing to psychological levels and reaching physical violence, to the point that all three types are eventually exercised simultaneously [9].

Numerous studies have highlighted the importance of family variables in the explanation of CPV. In this respect, most authors have focused on parental socialization styles [10,11,12]. However, no attention has been given to the role of communication, which is the channel through which family functioning and its quality is established. Thus, the correct adjustment of children depends largely on communication among family members [13,14]. Research on CPV and communication is in an incipient stage, hence the need for a more in-depth investigation and analysis of the relationships between these variables.

Through warm family communication, parents convey the importance of attending to emotions. Furthermore, positive and open communication in families helps to develop adolescents’ self-esteem and their emotional well-being [15]. However, critical, derogatory and punitive family communication practices imply a decrease in emotional regulation in minors. Consequently, in families where the parents make all decisions and the children act in accordance with their instructions, without discussing them, there is a decrease in emotional expressiveness and an increase in negative emotions [16].

Likewise, violent adolescents have greater difficulty in identifying, describing, understanding, accepting and regulating their emotions and expressing how they feel [17,18], in addition to experiencing negative emotions more frequently [19]. Therefore, they tend to feel misunderstood and use violence to handle situations that they are not emotionally capable of controlling [20,21]. Few studies have studied the relationship between emotional intelligence (EI) and CPV, hence the need for more in-depth research into this aspect. 

### 1.1. Family Communication and CPV

The relationships between adolescents and their parents influence CPV and the vehicle through which these relationships are organized is family communication. In families characterized by positive communication, where information flows freely and there is empathic understanding between parents and children, messages are transmitted clearly and precisely and both parents are consistent with each other, resulting in satisfactory interaction for both parties [22]. In families in which this positive communication prevails, the members avoid blaming others for conflicts. Each person assumes his or her own responsibility and is willing to admit mistakes and apologize, thus favoring the emotional, cognitive and social development of the children [23]. Additionally, both adolescents and parents who communicate positively report greater life satisfaction and better psychological adjustment [14,24].

In contrast, some families are characterized by negative communication, either because it is avoidant in nature and communication channels have been closed or because any communication that does take place is excessively problematic, critical and inefficient [22]. Negative communication with parents forms the basis of distrust towards adults [24] and is related to certain psychological adjustment problems in adolescents, such as violent behavior [25], criminal behavior [26] or peer aggression [27,28].

Likewise, some researchers have reported a two-way relationship between family communication and violent behaviors in children, whereby negative communication in the family precipitates the development of behavioral problems and, in turn, aggressive behavior in adolescents worsens communication with both parents. This relationship implies a decrease in the feeling of affective union within the family, which reacts negatively to the adolescent’s behavior and aggravates these communication difficulties [29,30,31,32].

More specifically, families affected by CPV are characterized by negative family climates, the absence of cohesion or closeness between members and scarce or poor communication skills. Parents are overly critical of their children’s actions and do not reinforce their positive behaviors [33]. Other studies have also found that adolescents who exercise CPV report greater exposure to aggressive and avoidant communication [4,12,34,35,36,37]. Additionally, CPV is associated (by children) with a lack of warmth when communicating with their parents [38]. Witnessing verbal aggression in the family is associated with verbal violence towards both parents [39], since witnessing communication based on insults, offences and coldness between the spouses makes children accustomed to this type of violence, which they internalize as an appropriate way to resolve conflicts.

However, fewer conflicts and few cases of CPV have been observed in families whose relationships are based on positive and open communication and affection, where there is greater expressiveness among family members [12,40]. 

### 1.2. EI and CPV

EI is the ability to perceive, value and express emotions accurately, to access and/or generate feelings in relation to thinking, to understand emotion and emotional knowledge and to regulate emotions in a way that promotes emotional and intellectual growth [41,42]. These authors developed an EI model, which they describe as comprising four dimensions: (a) the ability to perceive emotions, both their own and others, as well as musical, visual stimuli, etc.; (b) the assimilation or ability to generate, use and feel emotions to express feelings or influence cognitive processes; (c) emotional understanding or ability to understand emotions and their possible modifications or combinations; and (d) emotional repair or the ability to be open to feelings, as well as control and modify emotions to facilitate personal growth. Based on this model, EI is considered a core skill in information processing. The use of emotions allows us to think more intelligently, promoting more effective reasoning. Thus, emotions help solve problems and facilitate the adaptation of human beings to their environment. 

Emotional liability has been related with adjustment problems in adolescents. In young people, changes at brain level cause a predominance of emotions over cognitive components, slower maturation, which accounts for hypersensitive adolescents and their rapid changes in mood [43], as well as the tendency to blame others for emotional pain they cannot manage [44]. There is a greater association between EI and violence in adolescence than in adulthood as adults use more established cognitive programs to inhibit aggressive responses, while adolescents are still learning these guidelines and developing their emotional skills [45]. 

Studies in which EI is associated with violence in adolescents have reported conflicting results. Most research concludes that high levels of EI are related to better adjustment, less hostility and aggression [46,47], as well as less participation in criminal behavior [45,48,49]. It has been confirmed that, after training adolescents in EI, they learn to properly manage their emotions and significantly reduce the use of aggressive strategies to resolve interpersonal conflicts [50]. 

However, other authors have claimed that high levels of EI entail a greater predisposition for crime, as emotional manipulation requires a high capacity to understand emotions and feelings, as well as being able to predict how the victim will respond [51,52,53]. Due to this disparity in results, a more in-depth review is considered necessary. 

EI has also been related with the quality of family relationships, since one of the main reasons adolescents attack their parents is emotional in origin [54]. Some studies have concluded that adolescents who exercise CPV have greater difficulties identifying and expressing emotions, as well as interacting emotionally [5,12,55]. Other noteworthy emotional characteristics in these adolescents are hostility and the search for sensations [56], impulsiveness and difficulty controlling anger [57,58], as well as stress [59]. They also present low tolerance with frustration [60], low self-esteem [33] and lack of empathy [5,55,60]. 

Taking into account the importance of emotions management within the family context, as well as the link between EI and aggression and the difficulty in resolving conflicts adequately, EI is probably associated with CPV. A few studies have studied this relationship and report that adolescents who attack their parents have poorly developed EI [61,62]. For this reason, this study proposes incorporating EI as an explanatory variable of CPV and also considering gender differences.

### 1.3. The Role of Gender

Some studies have found that boys exercise more CPV [34,61] while others indicate that it is girls who use it most [52], and other researchers have not found any differences [9,60,63]. Due to this disparity of results, more in-depth research should be conducted into these interesting characteristics based on gender.

Girls attach more importance to communication with the family and define it as more frequent and empathetic, while boys talk about themselves in a less open way than girls [5]. Girls also excel in aggressive verbal behavior towards boys and tend to engage in threatening or insulting behaviors more often than boys, which is linked to an inefficient management of emotions in the case of girls [47]. 

However, some studies have concluded that greater family communication reduces emotional and behavioral problems only in girls [64]. Therefore, more in-depth research is considered essential.

Finally, according to the studies reviewed, girls obtain higher EI scores than boys [55,65]. Other authors have reported this same superiority in EI in women, with the added peculiarity that they have a lower self-perception of that EI than men [66]. Additionally, aggressiveness, in the case of boys, is more related to emotional distress than in girls [29]. 

### 1.4. The Present Study

The aim of this study is to analyse the relationship between CPV, family communication and EI in adolescents, but also taking into account the role of gender.

The following hypotheses are proposed:

**Hypotheses** **H1:**
*Teens with a high CPV score will obtain higher levels of offensive and avoidant communication and lower levels of EI than adolescents with medium and low levels of CPV.*


**Hypotheses** **H2:**
*An effect of interaction between CPV and gender will be observed, whereby children scoring high in CPV will obtain higher scores in offensive and avoidant communication and lower scores in EI than the other groups analysed.*


## 2. Materials and Methods

### 2.1. Participants

The empirical sample consisted of 1200 adolescents of both genders aged between 12 and 18 years (M = 13.88, SD = 1.32), enrolled at Compulsory Secondary Education (ESO), Baccalaureate and training centres in Andalusia and the Autonomous Community of Valencia. These areas were chosen because they are related to our research project. Of this sample, 611 were male (46.86%) and 693 were female (53.14%). The participants were selected by means of quota-based sampling according to the province, origin and socio-economic level of the area. The study was conducted at eight educational centres and the following criteria were considered: ownership (public: 80%—six centres—and private/state-subsidised: 20%—two centres); province; and origin (rural and urban).

### 2.2. Procedure

Firstly, a letter was sent to the management of the selected schools explaining the research project. About 5% of centers refused to participate for different reasons—2% did not participate because the staff declined the invitation, 3% of the schools were reluctant to participate and other centers rejected it because they had previously participated in other projects. Therefore, finally there were eight educational centers that participated. After confirming their interest and voluntary participation, an informative seminar was arranged with the teachers to explain the objectives and scope of the research to encourage their participation. Then, a letter explaining the research was sent to students’ families, requesting their written consent for their sons and daughters to participate in the study. After obtaining consent from the parents and students, the data were collected with each group in their regular classrooms during a fifty-five-minute session. All the teachers participated in the study on a voluntary, consensual and non-remunerated basis. Before their application, the aim of the research was briefly explained, the confidential and anonymous nature of their responses was guaranteed, as well as the voluntary nature of their participation and the possibility of abandoning the study at any time during the process. Subsequently, each participant was given a booklet with all the instruments together with instructions on the way to answer the tests. At least two members of the research team remained in the classroom to answer questions and ensure that the questionnaires were completed properly. Once the students finished completing the tests, they were handed over to the research staff, who put them in an envelope that was sealed in the presence of the students and on which the name of the centre, academic year and number of students in the classroom were noted down. The order of administration of the instruments was compensated in each classroom and school. This research was carried out in accordance with the ethical values required in research with human beings and took into account the fundamental principles included in the Declaration of Helsinki, as well as subsequent updates and current regulations on the right to information, informed consent, personal data protection, guarantees of confidentiality, non-discrimination and freedom to leave the study at any stage.

### 2.3. Instruments

*Child-to-parent violence*. The Child-to-Parent Aggression Questionnaire (CPAQ), adapted from Calvete et al. (2013) from the original by Straus and Douglas (2004), was applied. This instrument was used to measure physical and psychological violence towards fathers and mothers independently. The scale comprises 20 parallel items, 10 referring to the father and 10 to the mother, three of them measuring physical violence (e.g., hitting, kicking) and another seven measuring psychological violence (e.g., insulting, threatening, taking money without permission). The adolescents indicated how often they had carried out these actions against the father or mother in the last year using a four-point Likert scale: 0 (never), 1 (it has occurred once or twice), 2 (it has occurred between three and five times) and 3 (it has occurred six times or more). Cronbach’s alpha was 0.875 and the CFA showed a good fit of the model with the data for both mothers [SBχ2 = 48.3021, gl = 29, *p* < 0.05, CFI = 0.951, RMSEA = 0.022 (0.010, 0.033)] and fathers [SBχ2 = 41.4346, gl = 30, *p* = 0.07993, CFI = 0.959, RMSEA = 0.017 (0.000, 0.029)].

*Family communication*. The Parent–Adolescent Communication Scale (PACS) developed by Barnes and Olson (1982) was used, adapted by Estévez, Musitu and Herrero (2005), in order to measure three factors: open communication ("My father/mother listens to me"), offensive communication ("My father/mother insults me") and avoidant communication ("I am afraid to ask my father/mother what I want"). This scale consists of 20 items that provide information about the family communication style between parents and adolescent children on a response range from 1 (never) to 5 (always). Cronbach’s alpha was 0.936 (open communication), 0.782 (offensive communication) and 0.775 (avoidant communication). The model presented an acceptable fit with the data for both mothers [SBχ2 = 419.7378, gl = 144, *p* < 0.001, CFI = 0.943, RMSEA = 0.046 (0.041, 0.051)] and fathers [SBχ2 = 543.0522, gl = 142, *p* < 0.001, CFI = 0.946, RMSEA = 0.042 (0.041, 0.050)].

*Emotional intelligence*. The Perceived Emotional Intelligence Scale-24 (Trait Meta-Mood Scale-24), developed by Fernández-Berrocal, Extremera and Ramos (2004), adapted from the original TMMS-48 (Salovey, Mayer, Goldman, Turvey and Palafai, 1995), was used. This scale evaluates perceived intra-personal EI through three sub-scales: perceived emotional attention, i.e., attention to feelings or ability to feel and identify feelings properly ("I think about my mood constantly"); perceived emotional clarity, i.e., the ability to understand one’s emotional states ("I am often wrong about my feelings"); and perceived emotional repair, i.e., repair of moods or ability to control emotional states correctly ("Although I am sometimes sad, I have mostly an optimistic outlook"). All the items were written in a positive sense to facilitate understanding. When responding to the items, the subjects had to indicate their degree of agreement or disagreement using a five-point Likert scale (1 = disagree, 2 = somewhat agree, 3 = agree very much, 4 = strongly agree and 5 = totally agree). Its reliability—Cronbach’s alpha—was 0.91 (perceived emotional attention), 0.86 (perceived emotional clarity) and 0.97 (perceived emotional repair). The model presented a good fit with the data [SBχ2 = 707.0127, gl = 204, *p* < 0.001, CFI = 0.954, RMSEA = 0.043 (0.040, 0.047)].

### 2.4. Data Analysis

The independent variables were: CPV with three conditions—high (scores equal to or greater than the 75th percentile), medium (scores below the 75th percentile and greater than 25) and low (scores less than or equal to the 25th percentile); and gender, i.e., men and women. The dependent variables selected were offensive, avoidant or open family communication, and three variables related to EI, namely perceived emotional repair, perceived emotional attention and perceived emotional clarity.

A Multivariate Analysis of the Variance (MANOVA, 3 × 2) was performed in order to determine the differences in family communication and EI according to the involvement of adolescents in violent behaviors towards their parents. 

Then, ANOVA was performed to analyse the statistically significant differences in the variables and the Bonferroni post-hoc test was applied (α = 0.01).

## 3. Results

First, a cluster K-means analysis was performed to obtain the CPV groups. Three groups of adolescents were identified: low CPV (N = 201, 15.3%), moderate CPV (N = 793, 60.8%) and high CPV (N = 310, 23.8%). Table 1 shows the distribution of CPV in adolescents (low, moderate and high) according to gender (boy or girl). As can be observed, the percentage of boys and girls in each group was equivalent.

The MANOVA revealed statistically significant differences in the main effects of CPV [Λ = 0.858, *F* (12, 2586) = 17,127, *p* < 0.001, η_p_^2^ = 0.074] and gender [Λ = 0.969, *F* (6, 1293) = 6,875, *p* < 0.001, η_p_^2^ = 0.031]. No statistically significant interaction effect was observed between gender and CPV [Λ = 0.984, *F* (12, 2586) = 1,708, *p* = 0.059 η_p_^2^= 0.008].

With respect to CPV, the ANOVA showed significant differences in offensive communication [*F* (2, 1298) = 70.66, *p* < 0.001, η_p_^2^= 0.098], avoidant communication [*F* (2, 1298) = 68.39, *p* < 0.001, η_p_^2^ = 0.095], open communication [*F* (2, 1298) = 34.62, *p* < 0.001, η_p_^2^= 0.051] and perceived emotional repair [*F* (2, 1298) = 6.51, *p* < 0.01, η_p_^2^ = 0.010]. 

The Bonferroni test was applied to determine the minimum distances between the means in CPV that were significant, limiting the type I error rate to 1% in order to limit the alpha value to 0.01 and avoid increasing the type error I as a consequence of the dependence that might have existed between the different measures for the same subject. Significant differences were observed between the three CPV groups in offensive and avoidant communication, with the high CPV group presenting the highest levels of both types of communication compared to the other two CPV groups. Adolescents presenting moderate levels of CPV used both offensive and avoidant communicative styles to a greater extent than adolescents with low CPV scores. Significant differences were also found in the three CPV groups in relation to open communication. The low CPV group presented the highest levels of open communication, followed by the moderate CPV group, the lowest levels being observed in the high CPV group. As regards the significant differences in perceived emotional repair, the high CPV group presented the lowest levels of emotional repair compared to the other groups and adolescents with moderate levels of CPV presented lower perceived emotional repair than adolescents with low CPV. Table 2 shows means, standard deviation and ANOVA results between CPV (high, moderate and low) and variables communication (offensive, avoidant and open) and emotional intelligence (clarity, attention and repair).

In terms of gender, ANOVA revealed significant differences in offensive communication [*F* (1, 1298) = 4.58, *p* < 0.05, η_p_^2^ = 0.004], this type of communication being superior in the case of girls. As regards the EI variables, significant differences were found for emotional attention [*F* (1, 1298) = 30.05, *p* < 0.001, η_p_^2^ = 0.023], which was also superior in girls. Table 3 shows means, standard deviation and ANOVA results between gender and the variables communication (offensive, avoidant and open) and emotional intelligence (clarity, attention and repair).

## 4. Discussion

The aim of this study was to analyse the relationships between CPV, family communication and EI in adolescents, also considering gender differences. The results showed that adolescents with high CPV scores reported higher levels of offensive and avoidant communication than those who exercised low CPV. Additionally, low CPV levels were related to higher positive communication scores. These results coincide with those reported in previous studies associating poor or problematic communication with adolescents who exercise CPV [4,34,35,36,37,38], as well as with other studies that indicate that family relationships based on positive communication are associated with a lower incidence of CPV [12,40].

The teenagers who perceived communication with their parents as offensive or avoidant have probably internalized and become accustomed to these communicative styles in order to resolve conflicts. In this negative communication process, information is not transmitted clearly, emotions are not expressed and those of others are not understood. Likewise, there is also no place for the admission of mistakes or responsibility in confrontations. For adolescents lacking the communication skills necessary to express themselves adequately, aggression can be a more accessible and easier way of interacting with their parents, resulting in the deterioration of the family atmosphere [29,30,31,32]. However, adolescents who perceive positive communication with their parents are able to resolve conflicts and express their emotions more socially. These positive behaviors are reinforced by families and adolescents do not feel the need to express themselves negatively. In this connection, Patterson [67] highlights that communicative processes are bidirectional. Thus, when a child behaves negatively, the parents may try to mitigate or inhibit disruptive behavior through coercive discipline, which often implies conflictive or avoidant communication. These family interactions dominated by violent attitudes and behaviors are subsequently reproduced by children, who see them as a suitable model for resolving conflicts. 

As regards the formulated hypothesis, the authors also expected to find lower EI scores in the adolescents who exercised high levels of CPV. In this sense, significant results were only found with respect to perceived emotional repair, which was actually lower in cases of high CPV and higher in cases of low CPV. These results were consistent with those reported in studies associating lower EI with CPV [61,62], as well as with research relating high EI scores to greater hostility, aggression and crime [45,46,47,48,49,50]. However, the findings reported here contrast with those described in studies that associated high EI with criminal behavior [51,52,53]. These discrepancies may be due to the fact that the aforementioned studies focused on EI as a personality trait, whereas here it was treated as a mental ability, a form of intelligence linked to emotion [42]. The results obtained in this research highlight the importance of studying EI from a multidimensional perspective. Indeed, in this paper, only one component of EI—perceived emotional repair—was related to CPV.

Emotional repair is a resource for dealing with and controlling emotional states correctly. When teenagers with low scores in this variable experience negative emotions, they are unable to stop them and replace them with positive feelings. As a result, when they feel angry and misunderstood, they do not know how to manage those feelings in a socially appropriate manner and may attack their parents as a way of expressing their emotional discomfort and frustration. However, teenagers with higher emotional repair scores know how to overcome negative emotions by reassessing the situation and considering the most positive aspects in each case. This coping strategy is similar to Lazarus and Folkman’s Cognitive Evaluation Theory [68], which implies a positive reassessment of a context perceived as overwhelming and may explain the control of emotional responses to stressful circumstances. 

As regards gender differences, in the case of perceived emotional attention, the girls scored higher than the boys. These data suggest that girls make an effort to know and identify their emotions more than boys. This finding coincides with the results reported in studies in which girls obtained higher EI scores than boys [55,65]. Due to differences in gender socialization, girls are educated towards developing greater sensitivity towards their own and others’ emotions [66]. It is therefore consistent that they pay more attention to their emotions than boys.

Girls also scored higher than boys in the offensive communication variable, coinciding with the findings reported in other studies that conclude that girls obtained noteworthy scores in offensive communication [47], as well as studies that have described an increase in family communication and a decrease in behavioral problems in girls, but not in boys [64]. 

In some studies, it has been observed that girls show higher levels of family communication than boys, especially with their mothers [15]. This greater communication can also cause more conflictive interactions, typical of adolescence, related to the acquisition of greater autonomy and independence. However, girls, despite not agreeing with family restrictions that limit their freedom of conduct, obey their parents to a greater extent than boys [69]. These conflict situations in families with communication problems tend to be resolved unilaterally by parental imposition but, far from being solved, remain latent. This largely disputed form of conflict resolution is associated with greater family dissatisfaction, especially in girls, who are more sensitive to family conflicts than boys [70], potentially resulting in rumination, discomfort and recurring discussions with family. In short, communication is greater, but in times of crisis and adjustment, such as adolescence, it can also be more negative. Following Patterson’s coercion model [67], negative communication practices adopted by parents to impose their will may be reinforced, since they get their daughters to meet their demands. However, these practices also seem to prompt adolescents to employ this style of communication to impose their preferences or wishes on their parents. Thus, the behaviors of both parties in the communication process would be reinforced, resulting in a coercive feedback model. The results obtained in this study suggest that girls’ perception of the relationship with their parents, which is built primarily through communication, is related to the type of aggressive behavior they exercise [71]. Thus, adolescents who perceive communication with their parents as conflictive or avoidant tend to behave violently towards the latter. Parents and daughters should be encouraged to negotiate conflicts in order to achieve a consensus satisfactory for both parties, agreed through positive communication.

It is necessary to mention that this study had several limitations. It was based on information provided only by adolescents but could be complemented by also gathering the opinions of their families in order to study violent behaviors from different perspectives. It is also likely that some adolescents were not sincere about the aggression against their parents, because of shame or fear of being rejected, even though the questionnaires were voluntary, anonymous and confidential. The direct relationship between the different styles of family communication and the components of EI could also be examined. Future studies should analyse and compare CPV in rural and urban areas to examine differences in family relationships in both contexts. This study was based on a cross-sectional design in which no causality direction could be established. It would therefore be worthwhile conducting longitudinal studies for future research to identify the journeys of these adolescents. 

## 5. Conclusions

Given the significant increase in CPV in recent years, further research into the variables involved in the incidence of this behavior is essential. The results obtained in this research provide a greater insight into an issue of growing social concern. This research confirmed the importance of fostering adequate, positive and open communication in families, as this improves the quality of the family environment and reduces aggressive behavior. It is also important to highlight the relevance of promoting the development of EI from childhood, both in the family and in other contexts of special importance for minors, such as school, as it is related to the correct psychological adjustment of adolescents, all with a view to preventing future aggression problems inside and outside the family. CPV is a behavior that is difficult to observe and is often invisible, hence the need for collaboration between different professionals to prevent it. Adequate intervention programmes also need to be developed and implemented to foster the enrichment of communication in families, as well as the development of EI in adolescents.

## Figures and Tables

**Table 1 behavsci-09-00148-t001:** Distribution of CPV in adolescents according to gender.

CPV			Low	Moderate	High
Gender	Boys	N	104	384	123
		%	7.9	29.4	9.4
	Girls	N	97	409	187
		%	7.4	31.4	14.3
Total		N	201	793	310
		%	15.3	60.8	23.8

**Table 2 behavsci-09-00148-t002:** Means, standard deviation (SD) and ANOVA results between high, moderate and low levels of CPV and the offensive, avoidant and open communication, emotional clarity, emotional attention and emotional repair variables.

			CPV			
		High	Moderate	Low	F (2, 1298)	η^2^
Communication	Offensive	1.97 _a_ (0.69)	1.57 _b_ (0.49)	1.42 _c_ (0.51)	70.66 ***	0.098
	Avoidant	3.07 _a_ (0.74)	2.64 _b_ (0.74)	2.27 _c_ (0.79)	68.39 ***	0.095
	Open	3.50 _c_ (0.78)	3.82 _b_ (0.74)	4.06 _a_ (0.78)	34.62 ***	0.051
Emotional intelligence	Emotional clarity	3.29 (0.78)	3.35 (0.76)	3.43 (0.90)	1.59 ^ns^	0.002
	Emotional attention	3.36 (0.94)	3.22 (0.89)	3.29 (0.98)	1.20 ^ns^	0.002
	Emotional repair	3.27 _b_ (0.94)	3.40 (0.86)	3.57 _a_ (0.95)	6.51 **	0.010

Note: Mean (SD) * *p* < 0.05; ** *p* < 0.01; *** *p* < 0.001; ns = non-significative a > b > c.

**Table 3 behavsci-09-00148-t003:** Means, standard deviation (SD) and ANOVA results between gender and the offensive, avoidant and open communication, emotional clarity, emotional attention and emotional repair variables.

		GENDER			
		Boys	Girls	F (1, 1298)	η^2^
	Offensive	1.58 (0.55)	1.69 (0.60)	4.58 *	0.004
Communication	Avoidant	2.63 (0.77)	2.73 (0.79)	0.922 ^ns^	0.001
	Open	3.81 (0.79)	3.76 (0.76)	0.15 ^ns^	0.000
	Emotional clarity	3.34 (0.81)	3.35 (0.77)	0.08 ^ns^	0.000
Emotional intelligence	Emotional attention	3.10 (0.91)	3.41 (0.90)	30.05 ***	0.023
	Emotional repair	3.40 (0.89)	3.39 (0.89)	0.25 ^ns^	0.000

Note: Mean (SD) * *p* < 0.05; ** *p* < 0.01; *** *p* < 0.001; ns = non-significative.

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
