# Peer review of "The Role of Parental Communication and Emotional Intelligence in Child-to-Parent Violence"

_behavsci, 2019, doi:10.3390/bs9120148_

Round 1

Reviewer 1 Report

This is a welcome article for the journal and deals with an important theme for scholars, practitioners and policy makers. It is an interesting paper which sheds light on a growing social problem in Spain.

However, as it stands, the paper needs to address a number of issues to achieve publishable standard. These are explained below. 

Authors indicate that the number of complaints filed by parents against their children has dramatically increased in recent years. I suggest authors give a possible explanation to this porcentage raise. The method section of the paper explains that students were recruited from 8 different educational centres. Were more educational centres contacted and informed about the study? If so, which were the reasons for not taking part in the study? Also, one of the criteria stablished had to do with origin. Although it is not the main aim of the study, it could be interesting for future studies to analyze and compare rural and urban responses.

Reviewer 2 Report

 A very good presentation of the  study conducted by the authors.

Yet, for the reader to understand better the results of the present study, some adittional information should be provided, such as:

 General information about the regions from which the sample was chosen: do we have any information about CPV or teens violence in general in those areas? How these number are in comparison to the country's or literature numbers. These infos give to the reader a better understanding of the frame in which the study took place; and possibly may explain some of the results of the study.
